# Agent Clustering Strategy Based on Metabolic Flux Distribution and Transcriptome Expression for Novel Drug Development

**DOI:** 10.3390/biomedicines9111640

**Published:** 2021-11-08

**Authors:** Yao Ruan, Xiao-Hui Chen, Feng Jiang, Yan-Guang Liu, Xiao-Long Liang, Bo-Min Lv, Hong-Yu Zhang, Qing-Ye Zhang

**Affiliations:** Hubei Key Laboratory of Agricultural Bioinformatics, College of Informatics, Huazhong Agricultural University, Wuhan 430070, China; ry@webmail.hzau.edu.cn (Y.R.); 15812703067@163.com (X.-H.C.); jf980923@webmail.hzau.edu.cn (F.J.); lyg0929@webmail.hzau.edu.cn (Y.-G.L.); xlliang0409@gmail.com (X.-L.L.); lbm612@webmail.hzau.edu.cn (B.-M.L.); zhy630@mail.hzau.edu.cn (H.-Y.Z.)

**Keywords:** drug repositioning, genome-scale metabolic models, connectivity map, MoA

## Abstract

The network module-based method has been used for drug repositioning. The traditional drug repositioning method only uses the gene characteristics of the drug but ignores the drug-triggered metabolic changes. The metabolic network systematically characterizes the connection between genes, proteins, and metabolic reactions. The differential metabolic flux distribution, as drug metabolism characteristics, was employed to cluster the agents with similar MoAs (mechanism of action). In this study, agents with the same pharmacology were clustered into one group, and a total of 1309 agents from the CMap database were clustered into 98 groups based on differential metabolic flux distribution. Transcription factor (TF) enrichment analysis revealed the agents in the same group (such as group 7 and group 26) were confirmed to have similar MoAs. Through this agent clustering strategy, the candidate drugs which can inhibit (Japanese encephalitis virus) JEV infection were identified. This study provides new insights into drug repositioning and their MoAs.

## 1. Introduction

Computational biology has revealed that the expression pattern of cells can be related to a set of characteristics of genetic makeup. Therefore, the characteristics obtained from gene expression data can reflect clinicopathological phenotypes and drug effects [1]. In order to meet the challenges of high investment and low reward in drug development [2,3,4,5], computational approaches for matching transcriptomic signatures have been used for drug repurposing. By matching transcriptomic signatures, drug–disease comparisons [6] and drug–drug comparisons were conducted [1]. Both drug–disease association and drug–drug similarity approaches rely heavily on publicly available gene expression data. The connectivity map (CMap) comprised gene expression profiles generated by treating five cultured human cell lines with 1309 compounds [7]. Based on the CMap data, the method of matching transcriptomic signatures under various disease conditions has been successfully used to perform drug repurposing predictions. For instance, gene expression signature (GES) and CMap have been used as tools to repurpose drugs for bipolar disorder [8]. By searching for the CMap database, novel drugs for ovarian cancer based on recurrent gene signatures have been identified [9].

The premise of most CMap research is that the disease may be characterized by some important characteristic genes. The parameters for ‘signature genes’ may be influenced by multiple factors, such as population size, the cell lines under study, disease severity, or differential gene expression methods [10]. Meanwhile, gene interactions and gene function may be ignored in gene expression characteristics due to high correlation and the existence of a redundant gene [11]. Network biology methods have been used to resolve this issue. For example, a previous study has found two potential therapeutic compounds for gastric cancer through weighted gene co-expression network analysis (WGCNA) [1] and CMap analysis [12]. Through WGCNA of CMap, co-expressed gene modules with functional annotations have been identified. Drugs can be clustered based on module expression to infer their mechanism of action (MoA) [13]. In addition to gene modules, drug MoA modules can be established to reveal drug actions and perturbance of cellular systems [14]. In a previous study, a gene expression profile based on 1309 agent treatments is classified into 49 modules by FABIA (factor analysis for bicluster acquisition) [15]. The relationships between modules and transcription factors (TFs) reveal some new metabolic regulation mechanisms.

Compared to genes and proteins, metabolites are more closely related to the phenotype of an organism. Specifically, the health and disease states of the human can be described more accurately by the metabolic state of human cells, tissues, organs, and the organism [16]. The reconstruction of genome-scale metabolic models (GEMs) is one of the major modeling approaches for system-level metabolic studies. A GEM computationally describes a complete set of stoichiometry-based mass-balanced metabolic reactions in an organism through gene–protein-reaction (GPR) associations which are established based on genome annotation data and experimentally obtained information [17]. The GEM allows the prediction of metabolic flux values of an entire set of metabolic reactions using optimization techniques such as flux balance analysis (FBA) [18]. The GEM can be tailored into context-specific networks to implement various context-specific simulations by integrating omics data [19]. The context-specific GEMs have been used to explain metabolic malfunctions in cells under chronic or acute disease conditions and to identify effective therapeutic targets [20,21,22].

In view of the fact that the metabolic network reflects gene–protein-reaction relationships, metabolic flux distribution can more effectively reveal drug MoAs than gene modules, since drugs with similar metabolic characteristics may be in the same module. In this study, to explore common metabolic characteristics among the drugs in CMap, the gene expression profile was integrated into GEM. Then, 1309 agents were clustered based on metabolic flux distribution to obtain drug modules with similar metabolic characteristics, and the drug modules were used to infer MoAs. Compared to previous drug repositioning methods based on the CMap database, our method analyzes the similarity between drugs at a metabolic level, thus providing additional information on drug repositioning.

Japanese encephalitis virus (JEV) is a major cause of acute encephalopathy. It is transmitted by mosquitoes and is prevalent in Asian and Pacific areas [23]. No approved drug is available for the specific treatment of JEV infections. In order to address this and to verify the effectiveness of our method, a negative value of differential metabolic flux between JEV and healthy controls, was used for clustering 1309 agents in CMap to obtain the agent with potential therapeutic capabilities for JEV. Among the obtained agents, LY-294002, Tetrandrine, and Alsterpaullone have been reported as effective in inhibiting JEV [24,25,26]. In addition, the anti-JEV activity of Podophyllotoxin, Abamectin, Ticarcillin, and Streptozocin was verified by in vitro experiments. The clustering method developed in the present study provides a novel perspective for drug repositioning or MoAs analysis.

## 2. Materials and Methods

### 2.1. Data Acquisition

A human genome-scale metabolic model was downloaded as an sbml file from previous studies [27]. This model was an updated version of human metabolic model [28], which comprised 8079 reactions, 5588 metabolites, and 3665 genes.

Microarray data were downloaded from CMap [7], which comprised 7056 gene chips corresponding to five cultured human cell lines treated with 1309 agents. In 7056 gene chips, there were multiple gene expression levels of cells treated with different concentrations of the same agent, and the median value of these gene expression levels served as the representative expression value. At the same time, different drugs had different numbers of controls, and the mean expression value of genes treated with multiple controls was adopted. Finally, a total of 11,917 gene expression values were obtained under 1309 agents and 300 control treatments. These obtained gene expression values were subsequently used to reconstruct context-specific GEMs.

The RNA-seq data before and after JEV infection were available in the NCBI database under BioProject accession number PRJDB7761 (https://www.ncbi.nlm.nih.gov/bioproject/?term=PRJDB7761, accessed on 6 November 2021). There were four biological replicates of JEV-infected SH-SY5Y and the uninfected control. The transcript abundance (given as FPKM) of each gene was calculated using STAR [29] and RSEM software [30], and the average transcript abundance was used to reconstruct context-specific GEMs.

### 2.2. Reconstruction of Context-Specific GEM

A python library pyTARG (https://github.com/SergioBordel/pyTARG, accessed on 6 November 2021) [31] has been developed to automatically constrain human GEMs by imposing maximal boundaries on all the reactions in the model. This method allows computing metabolic flux distributions and evaluating the outcomes of restricting enzyme-catalyzed reaction rates; thus, it has been used to identify metabolic drug targets for cancer cell lines [27].

To create a context-specific metabolic model, the human genome-scale metabolic model has been constrained using the full-constraint function from pyTARG. Briefly, pyTARG sets the boundaries in a discrete way, which contributed to avoid numerical problems while performing linear optimization. For reversible reactions, the original lower and upper boundaries were set within the range of −1000~1000, while for irreversible reactions, the original constraint boundaries were set within 0~1000. The constrained boundary was the upper multiple of 10 of the gene expression value. Then, the metabolic flux distribution under different treatment conditions was calculated using FBA to obtain differential metabolic flux of each reaction between treatment group and control group.

### 2.3. Flux Balance Analysis

FBA is a linear program that simulates the maximum possible flux of the feasible flux distributions in context-specific GEMs [18]. The information of GEM can be condensed in the stoichiometric matrix, which contains the stoichiometric coefficients of each metabolite in each reaction. To solve the linear objective function (biomass production) and to calculate the flux of each reaction, the internal metabolites are considered to be in a stable state, which can be expressed in Equation (1):(1)$$S\cdot \vec{v}=\vec{0}\;v_{j}^{min}\leq v_{j}\leq v_{j}^{max}$$
where S represents the stoichiometric matrix; v is a vector of flux distribution of all reactions, and it is between lower and upper boundaries; $v_{j}$ is between $v_{j}^{min}$ and $v_{j}^{max}$. Each row in S represents a metabolite, and each column indicates a reaction. The prediction of metabolic flux distributions was conducted using the python library COBRApy (version 0.16.0 and solver glpk) [32].

### 2.4. Cluster Analysis

Since the number of characteristics is higher than the number of samples, the gene expression value matrix and metabolic flux distribution matrix have the curse of dimensionality. Therefore, scaling of these two matrices should be performed before principal component analysis (PCA) by using sklearn.preprocessing.MaxAbsScaler (https://scikit-learn.org/stable/modules/generated/sklearn.preprocessing.MaxAbsScaler.html, accessed on 6 November 2021) [33]. The PCA clustering was conducted by sklearn.decomposition. PCA(n_components=2) (https://scikit-learn.org/stable/modules/generated/sklearn.decomposition.PCA.html, accessed on 6 November 2021) [33].

The difference in metabolic flux between treatment group and the control group is defined as the metabolic characteristic of the cell. The similarity between the metabolic characteristics of cells is measured by Euclidean distance. This expressed by Equation (2):(2)$$d\left(x,y\right)\;=\sqrt{\sum_{i=1}^{n}{\left(x_{i}-y_{i}\right)}^{2}}$$
where n refers to the total number of reactions in GEM; $x_{i}$ is the flux of reactions i induced by agent *x*; $y_{i}$ is the flux of reactions i induced by agent y. The resultant matrix of agent metabolic flux similarities is then used to cluster agents with similar metabolic flux changes by using an affinity propagation algorithm [34] in sklearn.cluster.AffinityPropagation (https://scikit-learn.org/stable/modules/generated/sklearn.cluster.AffinityPropagation.html, accessed on 6 November 2021) [33] with default parameters.

The drug which has an opposite effect to that of a disease signature on transcription is considered to have the potential to ‘reverse’ the disease signature [1]. Differential metabolic flux was used as a characteristic of a drug or disease in this study. The drugs with a negative value of the differential metabolic flux induced by JEV was considered by those possessing the potential to treat JEV infection, and such a negative value was incorporated into the similarity matrix to cluster candidate drugs treating JEV.

### 2.5. Antiviral Activity Testing

BHK-21 cells were cultured in complete Dulbecco’s modified Eagle’s medium (DMEM) supplemented with 10% fetal bovine serum (FBS) as a growth medium or 2% FBS as maintenance medium.

JEV P3 strain used in this study was propagated in BHK-21 cells pregrown in 175 cm^2^ tissue culture flasks. Cells were infected with 100 μL virus stock, and then complete medium was added. Subsequently, cells were further incubated at 37 °C and 5% CO_2_. At the time of the highest cytopathic effects, the culture medium was frozen and thawed twice and centrifuged to remove cell debris. Afterwards, the supernatant was collected. The viruses were kept at −80 °C until use.

BHK-21 cells (1 × 10^4^ cells per well of a 96-well plate) were treated with various concentrations of each agent for 48 h, followed by a CCK-8 assay. Cell viability was calculated as the surviving cell number at each agent concentration divided by cell numbers in the control group (untreated with agent) at 450 nm (OD450). Each agent was applied in a concentration gradient, and the experiments were repeated for three biological replicates.

The cells were seeded in 12-well plates (1 × 10^5^ cells per well) for 24 h at 37 °C and 5% CO_2_. JEV culture supernatant was added to the plates at a multiplicity of infection (MOI) of 0.1, followed by incubation for two hours with gentle shaking every 15 min to achieve optimal virus-to-cell contact. The cells were washed twice with phosphate-buffered saline (PBS) after removing the virus culture supernatant. Unabsorbed virus was removed and different concentrations of agents (Podophyllotoxin: 0.12 μM, 0.06 μM, 0.03 μM; Ticarcillin: 0.12 μM, 0.06 μM, 0.03 μM; Streptozocin: 0.12 μM, 0.06 μM, 0.03 μM; Abamectin: 4.58 μM, 2.29 μM, 1.145 μM) or dimethyl sulfoxide (DMSO) vehicle control (1% DMSO) were added, and the cells were further incubated for 48 h. Viral titers were evaluated using plaque formation assays.

A 10-fold serial dilution solution of the culture supernatant of JEV-infected cells was added to fresh BHK-21 cells grown in 12-well plates (1.2 × 10^6^ cells) and incubated for 1.5 h at 37 °C. The cells were overlaid with DMEM containing 1.5% methylcellulose. Viral plaques were stained with crystal violet dye after a 3-day incubation period. Virus titers were calculated according to the following formula: titer (pfu/mL) = number of plaques × volume of the dilution solution × dilution folds of the virus solution.

BHK-21 cell line and JEV P3 strain were kindly provided by Prof. Yun-Feng Song, School of Animal Science and Technology, School of Animal Medicine, Huazhong Agricultural University, Wuhan, China.

## 3. Results

### 3.1. Reconstruction of Context-Specific GEMs

A genome-scale metabolic network of humans [27] was an updated version of the metabolic model of human metabolic reaction (HMR) series. The HMR series human metabolic model contains information on subcellular localization and tissue-specific gene expression mainly from the Human Protein Atlas database. Compared with the Recon series model, the HMR series model has more comprehensive information on fatty acid metabolism [35]. The microarrays from the CMap database [3] were used to impose maximal boundaries on all the reactions in the model by a full-constrain function from pyTARG [27]. A total of 7056 gene chips corresponding to five cultured human cell lines treated with 1309 agents were employed. In these gene chips, 2668 genes and 5597 reactions were mapped to the metabolic network model. A total of 1609 context-specific models were constructed by the above methods, 1309 models of which were those after agent treatment, and the remaining 300 models were controls (with no agent treatment) (Appendix A). In order to obtain the metabolic distribution of the context-specific GEMs, FBA of these models was performed. Because uncontrolled cell proliferation is one of the main characteristics of cancer, the objective function of FBA is to maximize the capability of producing cell biomass.

### 3.2. Metabolic Flux Serving as a Distinguishing Feature

Gene expression is often used as drug clustering characteristics to infer the MoA of a drug [13,15], but transcriptomes are complex data types with a high degree of heterogeneity; thus, biological signals may be obfuscated [36]. The context-specific GEMs have been used to identify effective therapeutic targets [20,21,22]. In order to determine whether the metabolic flux can distinguish the agent-treated cell from those untreated, the cells were clustered based on gene expression values and the metabolic flux distribution. The gene expression value matrix was comprised of 11,917 gene expression values and 1609 samples (Appendix A). These samples contained 1309 agent-treated and 300 untreated ones (control). Similarly, the metabolic flux matrix consisted of 8709 reaction fluxes and 1609 samples (containing 1309 agent-treated and 300 untreated ones). Since the number of characteristics (11,917 gene expression values and 8097 metabolic fluxes) was much higher than the number of samples (1609), PCA was performed [37] to improve the statistical significance and reduce noise caused by irrelevant features (Figure 1). Before PCA, the data were scaled to guarantee data within an appropriate range for model calibration [38].

Gene expression values were clustered by PCA, and the clustering results are shown in Figure 1a. In the feature space described by PC1 and PC2, the agent-treated cells (blue) and the untreated cells in the control group (red) were clustered in the lower left part and the middle and upper part with agent-treated cells and untreated cells overlapping, indicating that the gene expression values could not effectively distinguish agent-treated cells from those untreated.

The PCA results based on metabolic flux distribution are shown in Figure 1b. In the feature space described by the first two principal components based on flux clustering, the agent-treated cells (blue) were mainly clustered in three parts, and the untreated cells (red) were mainly distributed in the upper left part. The results showed that compared with gene expression values, metabolic flux distribution could better distinguish the cells before and after agent treatment. Although based on metabolic flux distribution, untreated cells were clustered together, with some drug-treated cells still overlapping with the untreated ones, indicating that some drugs induced metabolic flux distribution insufficient to distinguish the agent-treated cells from the untreated ones.

### 3.3. Cluster Analysis Based on Metabolic Flux Distribution

In order to discover the action mechanism of different agents on the cell lines, the agents with the differential flux distribution between the agent treatment and its control were further clustered by an affinity propagation algorithm [34]. Without requiring prespecification of the number of clusters, the affinity propagation algorithm used a collection of real-valued similarities between data points as input to group drugs with distinct metabolic profiles [39]. The metabolic characteristics of the cells were quantified as differential metabolic flux between the drug treatment and its control. The similarity of metabolic characteristics induced by different agents was used for clustering agents. Based on the characteristics of the differential metabolic flux data, the Euclidean distance was chosen to examine the similarity of the differential metabolic flux of different agents to implement the affinity propagation algorithm.

The results showed that 1309 agents were clustered into 98 groups, with the largest group containing 127 agents and 18 groups containing only one agent (Appendix A). The largest group containing 127 agents was related to numerous basic metabolic processes. The 18 smallest groups containing only one agent was related to no basic metabolic pathways. The largest and smallest groups were not further analyzed in the subsequent study. The remaining 79 groups containing 1165 agents were summarized and shown in Figure 2. Each remaining group contained about 15 agents on average. These 79 groups were considered to be able to reveal MOAs for further analysis.

### 3.4. Pharmacodynamics Analysis of Different Groups

Several strategies have successfully been used for computational drug identification. These strategies are based on the following principle that agents with same characteristics could have similar efficacy or action mechanisms [1]; thus, the known agents can be used to infer the potential pharmacology of other agents in the same group.

Group 26 mainly contained ten agents including antibiotics and neurological drugs, and the detailed information was shown in Table 1. All these agents in this group are approved by the Food and Drug Administration (FDA). Among these agents, Tacrine is a cholinesterase inhibitor to treat mild-to-moderate Alzheimer’s, and Tacrine can inhibit the metabolism of acetylcholine, thereby increasing its activity and raising its levels in the cerebral cortex [40]. Cisapride, belonging to group 26, is a 5-HT4 agonist developed for treating gastrointestinal disorders. Cisapride functions by stimulating the serotonin 5-HT4 receptors, thus increasing acetylcholine release in the enteric nervous system [41]. The 5-HT4R has been reported to be the promising biological target to prevent or slow down the progression of Alzheimer’s, and the 5-HT4 agonist can be developed to slow down Alzheimer’s pathology [42]. The above results confirm that the agents in the same group have similar MoAs.

Several previous studies have confirmed that as an antibiotic, Doxycycline has the potential to serve as a repositioned drug for Alzheimer’s disease (AD) [43,44,45]. In our study, Doxycycline was clustered into group 26. The above result showed that the metabolic network- based agent classification could divide the agents that could induce similar phenotypes of cells into the same group, which validates our analysis process. The above findings indicate that the pharmacology of unknown drugs can be inferred based on the pharmacology of known drugs, which provides new clues for drug reposition.

In addition to inferring the potential pharmacology of approved agents, grouping based on differential metabolic flux can also be used to explore the effects of agents unapproved by the FDA. Group 7 includes nine drugs, most of which are anti-inflammatory drugs. Eight of them have been approved by the FDA, one temporarily as an experimental drug, and the detailed information is shown in Table 2. In group 7, Tiaprofenic acid and Tenoxicam are known anti-inflammatory drugs, and other agents have not been used for anti-inflammation, but they all have potential anti-inflammatory effects. Some previous studies have revealed that in addition to lipid-lowering abilities, Lovastatin belonging to group 7 exhibits multiple anti-inflammatory effects [46,47]. Betahistine in group 7 has been reported to reduce the asymmetrical functioning of sensory vestibular organs and promote vestibulocochlear blood flow, thus decreasing symptoms of vertigo and balance disorders [48]. In addition, one previous study has reported that Betahistine can effectively suppress both inflammation and Th17 responses in mouse CIA, and that it may have therapeutic value as an adjunctant for rheumatoid arthritis [49]. Rifampin from group 7 is an antibiotic with a very broad spectrum of activity against most Gram-positive and Gram-negative bacteria (including *Pseudomonas aeruginosa*), specifically *Mycobacterium tuberculosis* [50]. Some studies have indicated that Rifampicin has anti-inflammatory effects [51,52].

In addition to the approved agents, experimental agent L-methionine sulfoximine was also clustered into group 7 with its pharmacodynamics unknown, and it has been reported to be a novel anti-inflammatory agent [53,54]. These results suggest that the drugs with a similar pharmacology from the same group may provide a reference of new drug development through this grouping approach. Further analysis of other groups (apart from group 7 and 26) can provide more insights into drug development.

### 3.5. MoA Prediction by Enrichment Information

To further explore the main metabolic mechanisms of the agents in each group, Metascape [55] was used to perform enrichment analysis of the genes associated with up/down-regulated reactions induced by agents in the 26th group and 7th group. For each agent in the same group, the genes with differential fluxes greater than 1 were selected for biological function analysis. GO and KEGG enrichment analyses indicated that in the 26th group, the up-regulated genes were mainly enriched in such KEGG pathways and GO terms as proton transmembrane transport, small molecule catabolic process, oxidative phosphorylation, carbon metabolism, cardiac muscle contraction, and pyruvate metabolism (GO:1902600, GO:0044282, hsa00190, hsa01200, and hsa04260, and hsa00620) (Appendix A). The down-regulated genes were mainly enriched in such GO terms and KEGG pathways as generation of precursor metabolites and energy, small molecule catabolic process, antibiotic metabolic process, carbon metabolism, nitrogen metabolism, and pyruvate metabolism (GO:0006091, GO:0044282, GO:0016999, hsa01200, hsa00910, and hsa00620) (Appendix A). These results showed that agent treatment greatly affects the energy metabolism of cells.

In the 7th group, the up-regulated genes were enriched in the following GO terms and KEGG pathways, such as proton transmembrane transport, small molecule catabolic process, organic anion transport, oxidative phosphorylation, and carbon metabolism (GO:1902600, GO:0044282, GO:0015711, hsa00190, andhsa01200) (Appendix A). The down-regulated genes were mainly enriched in such GO terms and KEGG pathways as the small molecule catabolic process, monocarboxylic acid catabolic process, organic anion transport, carbon metabolism, central carbon metabolism in cancer, and nitrogen metabolism (GO:0044282, GO:0072329, GO:0015711, hsa01200, hsa05230, and hsa00910) (Appendix A). These results indicated that agent treatment affects intracellular biosynthesis and metabolism.

Transcription factors (TFs) play a key role in signaling pathways, and they regulate many normal cellular processes; thus, they have been used as targets in disease [56]. Bio-related drugs have been reported to activate specific TFs [15]. In this study, to further analyze the biological function of the agents in the same group, Enrichr [57] was used to carry out TF enrichment analysis of the genes associated with up/down-regulation reactions induced by agents in the same group. Enrichr could provide 16 analysis results based on different data sources and analysis methods. In this study, the results of “ENCODE and ChEA Consensus TFs from ChIP-X” were chosen for further analysis.

Figure 3 showed the frequency of the top 20 enriched TFs of the up/down-regulated genes in group 7 and group 26. In group 7, transcription factor BHLHE40 of both up and down-regulated genes induced by the treatments of L-methionine Sulfoximine, Tenoxicam, Tiaprofenic acid, and Trimethadione was enriched (Figure 3a). BHLHE40, including Bhlhb2, Dec1, and Stra13, was up-regulated during T cell activation [58]. Yu et al. [59] have revealed that BHLHE40 serves as a molecular switch in maintaining the balance between the inflammatory cytokine IFN-γ and the anti-inflammatory cytokine IL-10 in Th1 cells both in vitro and in vivo. The TF MYC of down-regulated genes induced by the treatments of Lovastatin, Pyrazinamide, Rifampicin, and Tiaprofenic acid was enriched (Figure 3a). MYC plays an important role in re-programming the tumor microenvironment, especially the inflammatory and immune components of tumor stroma [60]. The TF KLF4 of the down-regulated genes induced by the treatments of Pyrazinamide, Betahistine, and Tenoxicam was enriched. The enriched KLF4 has been reported to be a proinflammatory factor because it activates epithelial cytokines in the esophageal squamous epithelium [61], and KLF4 in the colonic epithelium plays a crucial role in promoting DSS-induced colitis by modulating the NF-κB pathway inflammatory response [62]. The above results indicated that the agents in the 7th group activated TFs related to inflammation, which was consistent with their pharmacology.

The agents in the 26th group have pharmacology or potential pharmacology for the treatment of AD. The TFs enriched in the down-regulated genes seemed not to be associated with AD. The TF YY1 was enriched in up-regulated genes induced by nine agents (Figure 3b). YY1 plays a role in AD; thus, its overexpression increases the transcriptional activity of BACE1 (one of the major β-secretases) [63]. β and γ-secretase are the enzymes involved in the brain deposition of the amyloid-beta peptide (Aβ), which is one major hallmark of AD [64]. YY1 also might regulate the levels of Aβ indirectly by modulating the expression of other molecules (such as FE65) involved in APP processing. The expression level of FE65 in the brain of AD patients is related to the severity of the disease and the risk of developing delayed AD [65]. It has been reported that YY1 binds to the FE65 minimal promoter and increases its transcription [66]. The above findings and TF enrichment analysis results show that the agents in the same group might have similar MoAs. Notably, most of the enriched TFs are related to cancer, which is consistent with the fact that the cell lines in CMap are cancer cell lines.

Part of the TFs enriched from the 7th group were related to inflammation, while most of the TFs enriched from the 26th group were related to Alzheimer’s disease. The effects of these TFs are consistent with their pharmacology or potential pharmacology, indicating that the agents in the same group have similar MoAs. The above results suggest that the grouping method in this study can cluster the drugs with similar MoA. In addition to the above two groups (group 7 and 26), further analysis of other groups will provide more information on drug relocation and MoA.

### 3.6. Drug Repurposing for JEV Infection by Metabolic Flux Profile

To date, no approved drugs with activity against JEV infection are available. In order to identify potential drugs against JEV infection and to validate the effectiveness of our grouping method, the RNA-seq data (FPKM) before and after JEV infection were integrated into the metabolic network to obtain differential metabolic flux reflecting the metabolic characteristics of JEV. The metabolic characteristics of the agents against JEV were incorporated into the differential metabolic flux matrix of 1309 agents for clustering analysis. The clustered 31 agents were considered to be against JEV, of which 11 agents have been approved by the FDA (Appendix A). Table 3 showed the antibiotic and antiviral activity agents in this group consisting of these 31 agents. Among them, LY-294002 and Alsterpaullone, have been reported to effectively inhibit JEV, and LY294002 is a specific inhibitor of PI3K. Signaling pathway PI3K/AKT has been reported to be related to the infection process of the virus, and LY294002 blocks PI3K activation, thus greatly enhancing virus-induced apoptosis at an early stage of JEV infection [67]. Similarly, one study has revealed that JEV entry and internalization are impaired when PI3K activity is inhibited by LY294002 [24]. Alsterpaullone is a cyclin-dependent kinase (CDK) inhibitor, and this CDK inhibitor can inhibit the propagation of JEV by changing the JEV core protein morphology [25]. Tetrandrine contributes to the effective treatment of Ebola [68,69], and it has also been reported to significantly inhibit JEV infection [26]. These results supported that the agents with inhibitory effects on JEV infection have been successfully clustered into the group consisting of 31 agents.

The other agents in this group may have the potential to treat JEV infection. To determine whether the agent structures are similar, the similarity-based molecular fingerprints of agents were calculated using RDKit [72]. The results showed that the structure-based similarity between the agents in the anti-JEV infection group was very low (Appendix A). This suggests that structure-based similarity and metabolic flux distribution-based similarity reveal different aspects of molecular activity.

In recent years, antibiotics and phytochemicals have been considered as promising antivirals [71,73]. Some antibiotics and natural phytochemical agents in the anti-JEV infection group have also been confirmed to have antiviral effects, and Abamectin is found to have antiviral activity against alphaviruses and the yellow fever virus [70]. Podophyllotoxin belonging to this anti-JEV infection group is a lignan found in podophyllin resin from the roots of podophyllum plants. A lignan Arctigenin has been reported to have anti-JEV activity [74], but the activity of Podophyllotoxin anti-JEV infection is still uncertain. The above results suggest that the agents in this group have the potential to fight against JEV infection. In order to further verify the inhibitory activity of the agents in this group on JEV, Abamectin, Podophyllotoxin, and antibiotics (including Streptozocin and Ticarcillin based on their commercial availability) were selected for further anti-JEV infection experiments in vitro.

### 3.7. Antiviral Activity against JEV

Abamectin is a widely used insecticide and anthelmintic, and Podophyllotoxin is a natural phytochemical. Streptozocin and Ticarcillin are antibiotics. They belonged to the anti-JEV infection group in this study, and they were assumed to have the potential to fight against JEV infection. In order to further determine the effectiveness of our grouping, these four agents were selected for inhibition assay with viruses.

The cytotoxicity experiment indicated that the cell viability of the three agents (Podophyllotoxin, Streptozocin, and Ticarcillin) no longer increased when their concentrations were below 0.03 μM, and it was not less than 50% at all concentrations (Figure 4). The dose-dependent plaque reduction assay examined the inhibitory effects of these three agents on JEV at concentrations of 0.12 μM, 0.06 μM, and 0.03 μM, respectively. Similarly, Abamectin was less toxic to cells below the concentration of 4.58 μM. Thus, the concentrations of 4.58 μM, 2.29 μM, and 1.145 μM were chosen for the follow-up assay. The results of the plaque reduction assay showed that Podophyllotoxin inhibited JEV in a dose-dependent manner, which reduced JEV virus titer by 2 logs. Ticarcillin and Streptozocin had significant inhibitory effects on JEV (<1 log) (Figure 5b). High concentrations of Abamectin exhibited a strong inhibitory effect on JEV (Figure 5a). These results confirmed the inhibitory effects of the four agents on JEV in vitro, and these four agents were worthy of further study. Although the inhibitory activity of Abamectin was higher than that of Podophyllotoxin, Ticarcillin, and Streptozocin, and its concentration was much higher than the other three agents. The inhibitory effects of Podophyllotoxin, Ticarcillin, and Streptozocin on JEV at low concentrations indicated that these three agents had the potential to become new anti-JEV drugs. Overall, our results reveal that clustering strategies based on differential metabolic flux distribution can be used for identifying new drugs treating the diseases of interest.

## 4. Discussion

The present study aims to use the drug-induced metabolic distribution as a feature for drug clustering to identify novel drugs and reveal MoAs. The integration of genome-scale model and omics data allows us to obtain the reaction fluxes characterizing drug effects. Compared with the direct clustering method based on gene expression profiles, the metabolic flux distribution-based clustering method presented the relationship between genes, proteins, and reactions. In this study, 1309 agents from CMap database were grouped based on the GEM. Through our clustering method, the functions of unknown drugs could be inferred in terms of those of the known agents belonging to the same group. Undoubtedly, this greatly improved the efficiency of drug development and drug relocation. Some related studies have used the degree of negative correlation between the gene expression signature of the drug and that of a certain disease to infer whether drugs have a potential to treat this disease [1]. Similarly, our clustering method used the degree of negative correlation between the drug-induced metabolism flux and disease-induced metabolism flux to infer drug potential functions against JEV infection.

In this study, the 7th and 26th groups were selected to verify the effectiveness of our clustering method. The results showed that agents with similar pharmacology were successfully clustered into the same group. In addition, the GO and KEGG enrichment and TF enrichment analysis indicated that the agents in the same group had similar biological functions and MoAs. In the 68th group, there were six agents, three of which have been approved. Both Benzthiazide and Chloropyramine in group 68 can treat edema by inhibiting vasodilation [75,76]. More agent groups are suggested to be further analyzed in follow-up studies so as to reveal more drug MoAs.

Among the candidate drugs against JEV screened by our method, Abamectin and Podophyllotoxin showed good antiviral activity against JEV in vitro experiments. Ivermectin, a derivative of Abamectin, has been reported to inhibit replication of yellow fever virus and JEV by specifically targeting NS3 helicase [77,78]. Abamectin also has the potential to treat flavivirus [79]. Lignan has been confirmed to be a potential phytochemical targeting the JEV [80,81]. As a natural phytochemical, Podophyllotoxin is less toxic, and it is less likely for the virus to develop drug resistance [72]. Ticarcillin and Streptozocin, as antibiotics, also exhibited inhibitory activity against JEV. All of the above four agents can be studied further to develop anti-JEV drugs.

However, the number of agents in the CMap database is not enough, and these agents are only used to treat cancer cell lines, thus limiting the scope of drug development or drug relocation. With the increase in available data, future work can attempt to integrate the data with different sources and to comprehensively reveal drug MoAs. Further studies are suggested to examine the antagonistic or synergistic effects of drugs based on metabolic flux.

## 5. Conclusions

The CMap database is commonly used to identify novel drugs or reveal MoAs. However, only using a set of gene characteristics to predict drug functions and reveal drug MoAs may result in the ignorance of the gene function redundancy and gene–gene interactions. In this study, GEM was used to integrate the gene expression values of 1309 agents to obtain metabolic flux distribution. Based on the system-level metabolic distribution, the agents were clustered to identify the drugs with similar MoA. The literature confirms that the agents in the same group have the potential pharmacology. Enrichment analysis reveals that the TFs enriched in the up/down-regulated genes induced by agents in the same group have similar functions, which helps to analyze the MoAs of the agents in the same group. Our drug clustering strategy can be used to develop new drugs against JEV infection, and our results are supported by in vitro experiments. This study provides new insights into drug repositioning and their MoAs.

## Figures and Tables

**Figure 1 biomedicines-09-01640-f001:**
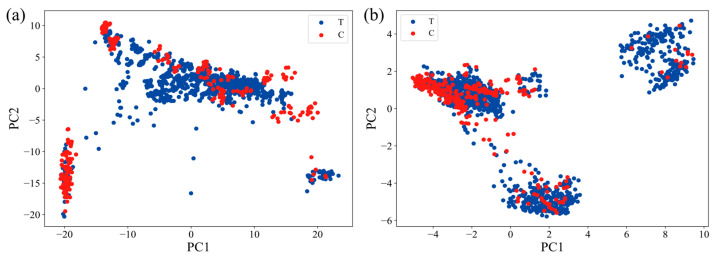
(**a**) PCA plot of 1309 agents based on gene expression. (**b**) PCA plot of 1309 agents based on metabolic flux. Blue and red, respectively, indicate agent-treated and control.

**Figure 2 biomedicines-09-01640-f002:**
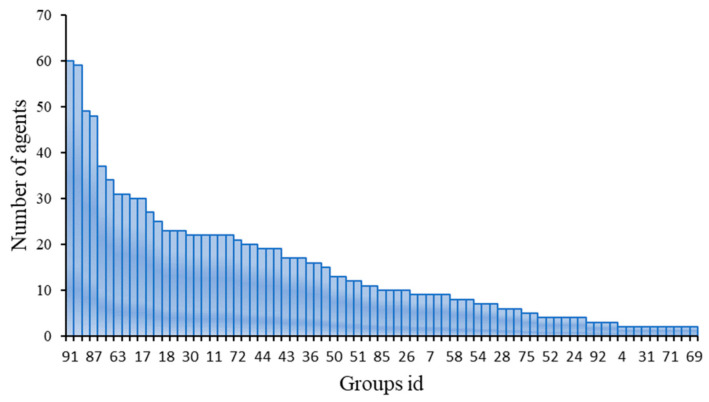
The number of agents contained in different groups.

**Figure 3 biomedicines-09-01640-f003:**
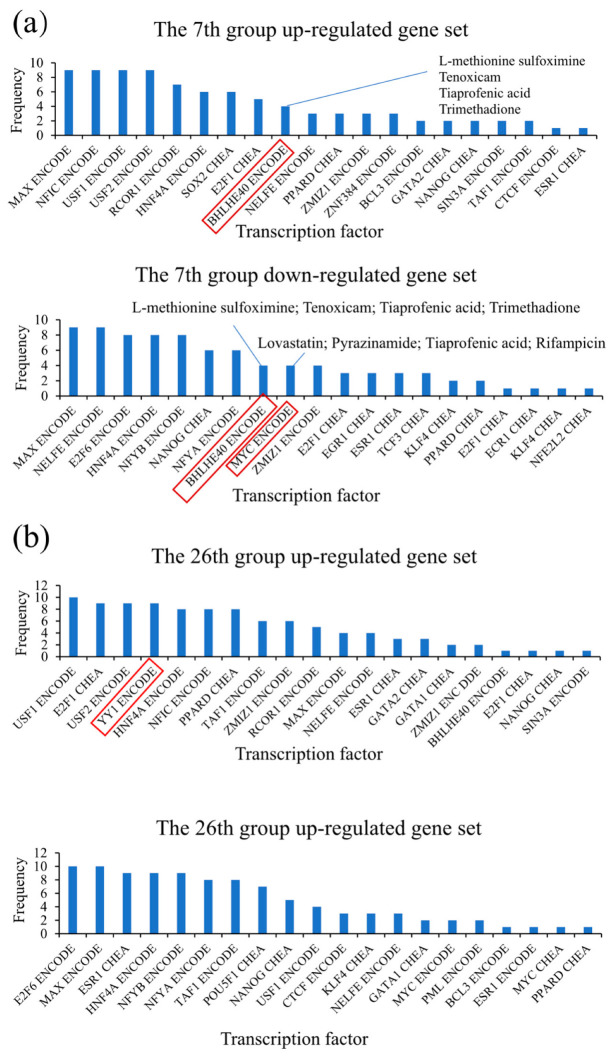
The TFs’ frequency of occurrence in groups 7 and 26. (**a**) The frequency of the top 20 enriched TFs of the up/down-regulated genes in the group 7. (**b**) The frequency of the top 20 enriched TFs of the up/down-regulated genes in the group 26.

**Figure 4 biomedicines-09-01640-f004:**
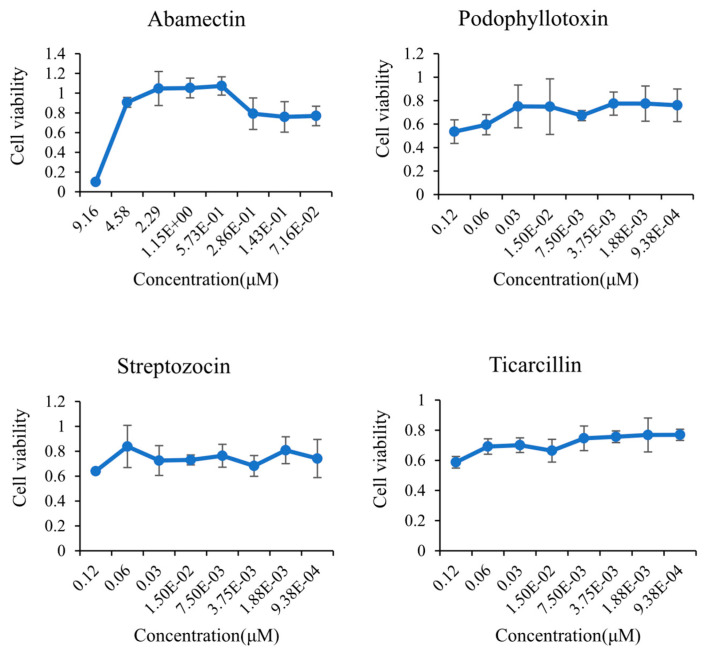
The cytotoxicity test of Abamectin and Podophyllotoxin, Streptozocin, and Ticarcillin at different concentrations.

**Figure 5 biomedicines-09-01640-f005:**
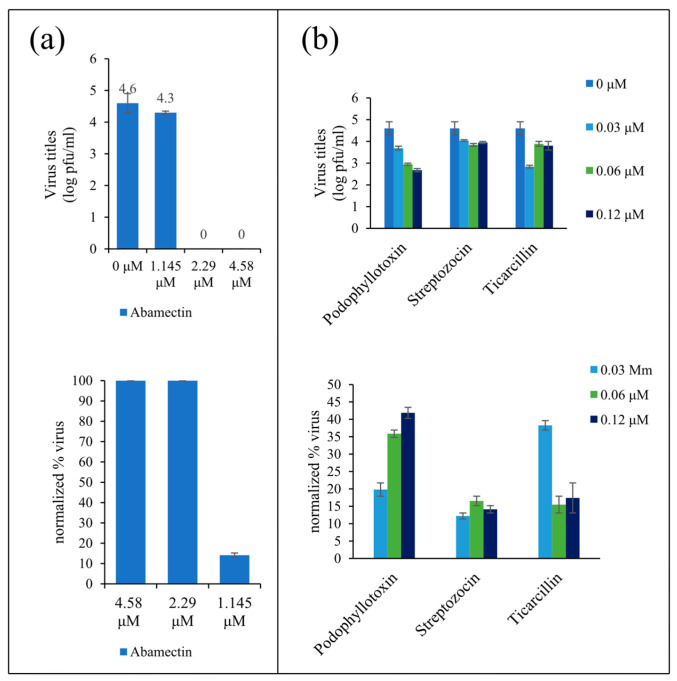
The effect of four agents on virus titer. (**a**) The JEV titers and inhibition rate on JEV in the presence of different concentrations of Abamectin. (**b**) The JEV titers and inhibition rate of various agents on JEV. Data are mean ± standard deviation of three independent replicates.

**Table 1 biomedicines-09-01640-t001:** The agent information of group 26.

Name	State	Pharmacodynamics
Doxycycline	Approved	antimicrobial
Nitrofural	Approved	antimicrobial
Sulfanilamide	Approved	antimicrobial
Ceforanide	Approved	antimicrobial
Dicloxacillin	Approved	antimicrobial
Neostigmine	Approved	cholinesterase inhibitor
Tacrine	Approved	cholinesterase inhibitor
Velnacrine	Approved	cholinesterase inhibitor
Cisapride	Approved	parasympathomimetic agent
Pilocarpine	Approved	parasympathomimetic agent

**Table 2 biomedicines-09-01640-t002:** The agent information of group 7.

Name	State	Pharmacodynamics	Potential Efficacy
Lovastatin	Approved	oral antilipemic agent	anti-inflammatory [46,47]
Trimethadione	Approved	anticonvulsant	-
Tiaprofenic acid	Approved	anti-inflammatory	-
Tenoxicam	Approved	anti-inflammatory	-
Terconazole	Approved	antifungal	-
Betahistine	Approved	histamine H1-agonist	rheumatoid arthritis [49]
Rifampicin	Approved	antibiotic	anti-inflammatory [51,52]
Pyrazinamide	Approved	antimicrobial	-
L-methionine sulfoximine	Experimental	-	anti-inflammatory [53,54]

**Table 3 biomedicines-09-01640-t003:** The agent information of the anti-JEV group.

Name	State	Pharmacodynamics	Anti-Virus
LY294002	Experimental	not available	JEV [24]
Alsterpaullone	Experimental	not available	JEV [25]
Abamectin	-	insecticide	Flavivirus [70]
Viomycin	Approved	antibiotic	-
Podophyllotoxin	Approved	lignan	Flavivirus [71]
Streptozocin	Approved	antibiotic	-
Tetrandrine	Experimental	not available	JEV [26]
Ticarcillin	Approved	antibiotic	-
Cefmetazole	Approved	antibiotic	-
Atovaquone	Approved	antibiotic	-

## Data Availability

Gene expression datasets are publicly available (CMap: available https://portals.broadinstitute.org/cmap/, accessed on 6 November 2021; NCBI BioProject: https://www.ncbi.nlm.nih.gov/bioproject/?term=PRJDB776/, accessed on 6 November 2021. The HMR model and pyTARG can be found at: https://github.com/SergioBordel/pyTARG, accessed on 6 November 2021.

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
