# Peer review of "Agent Clustering Strategy Based on Metabolic Flux Distribution and Transcriptome Expression for Novel Drug Development"

_biomedicines, 2021, doi:10.3390/biomedicines9111640_

Round 1

Reviewer 1 Report

The manuscript "Agent clustering strategy based on metabolic flux distribution and transcriptome expression for novel drug development" by Y. Ruan et al. presents a new clustering method to identify drugs with similar mechanisms of action (MoA) using the drug-induced metabolic distribution as a feature. The new method was employed in inferring the MoA of unknown drugs from knows agents that belong to the same cluster with improved efficiency compared with direct clustering methods based on gene expression profiles. The authors analysed the results of 2 groups and verified the effectiveness of the new clustering method in identifying drugs with similar pharmacological profile.

The work presented in this manuscript is novel and its applicability is immediate, thus I believe it will be of interest to the readers of Biomedicines. The quality of presentation is very good and so is the presentation of the methods in such detail to be easily reproduced. Overall, the analyses were carried out with statistically sound methods and the experiments were performed with standard protocols. Importantly, the conclusions presented by the authors are supported by the results and this work may boost further drug repositioning efforts.

Therefore, I suggest its publication to Biomedicines after a thorough proof-reading to correct some minor grammatical issues.

Author Response

We sincerely thank the reviewer for reading our paper carefully and giving the above positive comments. Your suggestions are very helpful for us to improve our work. The revised manuscript has been thoroughly checked and the grammatical errors and typos have been corrected (Line 231 of page 5 and Line 388-393 of page 10). We have revised the manuscript (presented by using the “track changes” mode in MS Word) and submitted it to the journal again.

Reviewer 2 Report

In the present manuscript authors approached metabolic network to connect different drugs with similar mechanism of action to develop new therapeutic strategies for pathological conditions. They identified new therapeutic drugs using this approach for Japanese encephalitis virus (JEV). They further validated their observation using model cell lines. The manuscript is well written and this study provides new avenues to search for treatments of other diseases. However, the manuscript needs experiments to validate effects of drugs.

Major comment:

  1. 4 cytotoxicity needs to be validated with cell cycle / or apoptosis markers to validate the graphs.

Minor comment:

  1. 5a is lacking the graphs for 2.29 and 4.58 µM. Authors need to include it.

Author Response

Sincere thanks should be given to the reviewer for the constructive comments and suggestions. We have tried our best to revise the manuscript according to your kind and construction comments and suggestions (presented by using the “track changes” mode in MS Word) and submit it to the journal again.

The point-to-point responses to reviewers’ questions are listed as below.

Major comment:

4 cytotoxicity needs to be validated with cell cycle / or apoptosis markers to validate the graphs.

Reply:

Thank you for your constructive suggestions, we understand that cell cycle/apoptosis markers may better validate the graphs. However, in the present study, we mainly focused on whether the drugs were identified have the potential to inhibit JEV. And we think that the CCK-8 assay may not be optimal, but should be sufficient to determine the concentration range. Zhang et.al found that the number of apoptotic cells increased significantly after being treated with 0.75 μmol/L podophyllotoxin for 24 h [1]. Jiang et.al. have proved that 0.01 μM podophyllotoxin could induce early apoptosis in porcine oocytes[2]. Zhu et.al. found that Abamectin blocks the cell cycle and induces apoptosis in MGC803 cells(0, 1, 2, 3, and 4 μM for 24 hours)[3]. We have not found any literature about the effects of Streptozocin and Ticarcillin on cell cycle or apoptosis. However, more experiments on the screened four agents are under investigation in our laboratory, the results will be announced in the future study.

Thanks again to the reviewer for suggesting to further improve this manuscript.

Minor comment:

5a is lacking the graphs for 2.29 and 4.58 µM. Authors need to include it.

Reply:

Thank you for your suggestion. I'm sorry that our pictures are misleading. Abamectin has a strong inhibitory effect on JEV at concentrations of 2.29μM and 4.58μM (almost 100%). This is reflected in the figure in the lower part of 5a. We think your suggestion is very important, so we made a slight modification to 5a (the concentration is marked on the abscissa and the data label added).

Thanks again to the reviewer for suggesting how to further improve this manuscript. We have studied comments carefully and have made corresponding corrections which we hope can meet with approval.

References

  1. Zhang, W.; Liu, C.; Li, J.; Liu, R.; Zhuang, J.; Feng, F.; Yao, Y.; Sun, C. Target analysis and mechanism of podophyllotoxin in the treatment of triple-negative breast cancer. Frontiers in Pharmacology 2020, 11, 1211.
  2. Jiang, W.-J.; Hu, L.-L.; Ren, Y.-P.; Lu, X.; Luo, X.-Q.; Li, Y.-H.; Xu, Y.-N. Podophyllotoxin affects porcine oocyte maturation by inducing oxidative stress-mediated early apoptosis. Toxicon 2020, 176, 15-20.
  3. Zhu, S.; Zhou, J.; Zhou, Z.; Zhu, Q. Abamectin induces apoptosis and autophagy by inhibiting reactive oxygen species‐mediated PI3K/AKT signaling in MGC803 cells. Journal of biochemical and molecular toxicology 2019, 33, e22336.

Round 2

Reviewer 2 Report

In the present manuscript, authors carefully addressed all comments and modified manuscript accordingly. I feel strongly the present form of manuscript is suitable for publication.